Perceptual elements in Penn & Teller’s “Cups and Balls” magic trick

Rieiro Hector 1 2 3
Martinez-Conde Susana 2
Macknik Stephen L. macknik@neuralcorrelate.com 1 2
1 Department of Neurosurgery , Barrow Neurological Institute , United States
2 Department of Neurobiology , Barrow Neurological Institute , United States
3 Department of Signal Theory and Communications , University of Vigo , Spain
Reser David
Electronic publication date: 2013 Feb 12
Publication date: 2013
Volume: 1
Electronic Location ID: e19
Received 2012 Dec 1; Accepted 2013 Jan 5
Copyright: © 2013 Rieiro et al.
Copyright year: 2013
Copyright holder: Rieiro et al.
License: This is an open access article distributed under the terms of the Creative Commons Attribution License, which permits unrestricted use, distribution, and reproduction in any medium, provided the original author and source are credited.
License URL: https://creativecommons.org/licenses/by/3.0/

Keywords: Sleight of hand, Magician, Social misdirection, Joint attention, Inattentional blindness

Funding: National Science Foundation 0726113 0852636 1153786 This work was funded by awards from the Barrow Neurological Foundation to SLM and SM-C, and from the National Science Foundation to SLM and SM-C. HR was a fellow of Fundacion Ibercaja. The funders had no role in study design, data collection and analysis, decision to publish, or preparation of the manuscript.

==============================
Magic illusions provide the perceptual and cognitive scientist with a toolbox of experimental manipulations and testable hypotheses about the building blocks of conscious experience. Here we studied several sleight-of-hand manipulations in the performance of the classic “Cups and Balls” magic trick (where balls appear and disappear inside upside-down opaque cups). We examined a version inspired by the entertainment duo Penn & Teller, conducted with three opaque and subsequently with three transparent cups. Magician Teller used his right hand to load (i.e. introduce surreptitiously) a small ball inside each of two upside-down cups, one at a time, while using his left hand to remove a different ball from the upside-down bottom of the cup. The sleight at the third cup involved one of six manipulations: (a) standard maneuver, (b) standard maneuver without a third ball, (c) ball placed on the table, (d) ball lifted, (e) ball dropped to the floor, and (f) ball stuck to the cup. Seven subjects watched the videos of the performances while reporting, via button press, whenever balls were removed from the cups/table (button “1”) or placed inside the cups/on the table (button “2”). Subjects’ perception was more accurate with transparent than with opaque cups. Perceptual performance was worse for the conditions where the ball was placed on the table, or stuck to the cup, than for the standard maneuver. The condition in which the ball was lifted displaced the subjects’ gaze position the most, whereas the condition in which there was no ball caused the smallest gaze displacement. Training improved the subjects’ perceptual performance. Occlusion of the magician’s face did not affect the subjects’ perception, suggesting that gaze misdirection does not play a strong role in the Cups and Balls illusion. Our results have implications for how to optimize the performance of this classic magic trick, and for the types of hand and object motion that maximize magic misdirection.

Introduction

Magic is one of the oldest art forms, and magicians have manipulated audiences’ perception and cognition for much longer than cognitive scientists have (Martinez-Conde & Macknik, 2007; Martinez-Conde & Macknik, 2008; Macknik et al., 2008). Thus, classic and contemporary magic illusions provide scientists with methodological refinements and testable hypotheses about the building blocks of conscious experience (Cui et al., 2011; Otero-Millan et al., 2011). The “Cups and Balls” is a sleight-of-hand magic trick that was performed by Roman conjurers as far back as two thousand years ago (Christopher & Christopher, 2006). The trick has many variations, but the most common one uses three balls and three cups. The magician makes the balls pass through the bottom of cups, jump from cup to cup, disappear from a cup and turn up elsewhere, turn into other objects, and so on. The cups are usually opaque and the balls brightly colored. Here we examined a version of this trick inspired by a routine performed by the entertainment duo Penn & Teller, conducted with three opaque and subsequently with three transparent cups.

Magician Teller devised this variation while fiddling with an empty water glass and wadded-up paper napkins for balls, at a Midwestern diner (Macknik, Martinez-Conde & Blakeslee, 2010). He turned the glass upside down and put a ball on top, then tilted the glass so that the ball fell into his other hand. The falling ball was so compelling that it even drew his own attention away from his other hand, which was deftly and automatically loading a second ball under the glass (he was so well practiced that he no longer needed to consciously control his hands). In fact, Teller found that the sleight happened so quickly he himself did not realize he had loaded the transparent cup. Teller further realized that all of this took place despite the fact that he should have been able to see the secret ball as it was loaded under the cup. Its image was on his retina, but he nevertheless missed it because his attention was so enthralled with the falling ball. He surmised that if it worked for him with a transparent cup, it would work with an audience. The transparency of the cups would make the trick all the more magical to the audience. Penn & Teller claim that their version of the trick violates four rules of magic: don’t tell the audience how the trick is done, don’t perform the same trick twice, don’t show the audience the secret preparation, and never perform cups and balls with clear plastic cups.

Here we set up to investigate whether the falling ball in Penn & Teller’s “Cups and Balls” generated stronger misdirection, as hypothesized by Teller, than alternative manipulations. Teller used his right hand to load (i.e. introduce surreptitiously) a small ball inside each of two upside-down cups, one at a time, while using his left hand to remove a different ball from the upside-down bottom of the cup. The third cup sleight involved one of six manipulations: (a) standard maneuver (i.e. ball falling to the magicians’ hand), (b) standard maneuver without a third ball, (c) ball placed on the table before going to the magician’s pocket, (d) ball lifted before going to the pocket, (e) ball dropped to the floor, and (f) ball stuck to the cup. See Supplemental Movies S1-6. Seven subjects watched the videos of the performances while reporting, via button press, whenever balls were removed from the cups/table (button “1”) or placed inside the cups/on the table (button “2”).

Subjects’ perception was more accurate with transparent than with opaque cups. Perceptual performance was worse for the conditions where the ball was placed on the table, or stuck to the cup, than for the standard maneuver. The condition in which the ball was lifted displaced the subjects’ gaze position the most, whereas the condition in which there was no ball caused the smallest gaze displacement. Thus, neither the standard falling ball or the enhanced falling ball condition (where the ball fell to the floor) generated the strongest misdirection, either in terms of perceptual performance or gaze position, contrary to the magician’s expectation.

Training improved the subjects’ perceptual performance. Occlusion of the magician’s face did not affect the subjects’ perception, suggesting that gaze misdirection does not play a strong role in the “Cups and Balls” illusion. Our results have implications for how to optimize the performance of this classic magic trick, and for the types of hand and object motion that maximize magic misdirection.

Methods

Subjects

Seven naive subjects participated in the experiment. All participants had normal or corrected-to-normal vision, and were paid $15 dollars for a single experimental session. The experiment was carried out under the guidelines of the Barrow Neurological Institute’s Internal Review Board (protocol 04BN039), and written informed consent was obtained from each participant.

Eye movement recordings

During the experiment, subjects rested their head on a chin/forehead-rest 57 cm away from a video monitor (Barco Reference Calibrator V), while free viewing the video clips. Their eye movements were non-invasively recorded with a video-based eye tracker (Eyelink 1000, SR Research), at 500 samples per second. From the eye tracker recordings, we identified and removed blink periods as the portions of the recorded data where the pupil information was missing. Furthermore, we removed the 200 ms before and after each identified blink period, to eliminate periods of time in which the pupil is partially occluded.

We identified saccades using an objective algorithm (Engbert & Kliegl, 2003). To reduce spurious positives due to noise, we analyzed only binocular saccades (i.e. saccades with at least one sample of overlap in both eyes). Furthermore, we ensured that overshoot corrections were not counted as saccades by imposing a minimum intersaccadic interval of 20 ms (Otero-Millan et al., 2011).

Experimental design

Subjects sat in a dark, quiet room and watched video clips of 10 to 12 s each, in which Teller performed different variations of a “Cups and balls” magic routine. The videos had a resolution of 720 × 480 pixels and subtended an area of 28 × 19 degrees of visual angle inside the visual field. The average luminance of the clips was 23 cd/m2, and their contrast ratio (full on/full off) was 128:1. Areas of the screen not occupied by the video were white.

In each clip, Teller performed the manipulation sequentially in each of three different cups, located from left to right on the screen. The manipulation in the first two cups was identical in all the clips (“Standard” load, see below), whereas the routine used in the third cup varied in each video clip (Fig. 1). After the third cup’s sleight was complete, Teller individually lifted all three cups to show the balls hidden underneath them. Subjects were instructed to report, as fast as possible, the removal and placing of each ball as soon as they were aware of them, by pressing one of two different buttons on a gamepad with their left and right index fingers (button “1” for removals, button “2” for placings, see Fig. 2). A removal was defined as the moment each ball stopped touching either the table or a cup, and a placing was defined as when each ball made physical contact with a cup or the table.

Figure 1 Summary of the different magic routines tested in the experiment.

In “Standard”, the routine performed on the third cup was identical to that performed for the previous two cups. In “No ball”, the routine was again the same as in the first two cups, but there was no ball initially placed on top of the cup. In “Lift”, the ball initially on top of the third cup was lifted to approximately eye level before the cup was loaded. In “Table”, the ball originally on top of the third cup was placed on the table before the cup was loaded. In “Drop”, the ball was dropped out of the screen before the third cup was loaded. In “Stuck”, the ball was attached to the top of the third cup. (Courtesy of NOVA scienceNOW/WGBH.)

Figure 2 Example trial.

Schematic of a single trial for the “Standard” routine. The spikes in the time courses represent actual (dashed lines) and reported (solid lines) loads and removals. Blue symbolizes removals and red placings.

The different routines tested were: (a) Standard (Supplemental Movie S1): the standard maneuver, identical to the one performed in the first two cups.

(b) No ball (Supplemental Movie S2): similar to the “Standard” routine, but there was no ball on top of the third cup.

(c) Lift (Supplemental Movie S3): the ball on top of the third cup was lifted to eye level before loading the cup.

(d) Table (Supplemental Movie S4): the top ball on the third cup was placed on the table before the cup was loaded.

(e) Drop (Supplemental Movie S5): the top ball on the third cup was dropped to the floor before the cup was loaded.

(f) Stuck (Supplemental Movie S6): similar to the “Drop” condition, but the ball was stuck to the cup and therefore it did not fall.

For each of these different routines, we tested other variables concerning the magician’s performance. We tested “Clear cups”, in which the cups were transparent, versus “Opaque cups”, in which they were not. We also tested “Load” versus “No load” conditions, in which the third cup was either “loaded” (i.e. a ball was surreptitiously placed under it), or not. Finally, we compared a “No face” condition, in which the magician’s face was occluded by a static black rectangle, versus the unmodified “Face”-visible video clips. This yielded a total of 48 conditions. Each subject saw each condition twice. The order of conditions was blocked and randomized for each subject. Each participant saw all the 48 conditions first in random order, and then the same conditions again in a new random sequence.

Data analysis

We defined a correct report of ball placing or removal as an appropriate button press in the 2000 ms immediately following the first movie frame in which the ball had been placed or removed. We also coded correct reports when subject did not indicate a placing after the magician performed a faked load. The reaction time of each report was measured in the conditions in which the cup was loaded. For each placing, gaze distance was calculated as the average distance between the subjects’ gaze and the point where the cup sat on the table, during the 400 ms immediately subsequent to, and following, the first movie frame in which the load occurred (or the equivalent frame in the “No load” condition). We varied the duration of these two time windows and found that the results were similar. Subjects were allowed to report ball placings during the reveal sequence at the end of each trial, in which the magician lifted the cups to show their contents. We counted the number of reports the subjects made during this period in each trial and considered them “late findings”.

Statistical testing employed a logistic regression fit to correct reports of placings and removals, and a linear regression fit to the reaction times and the gaze distances. The different magic routines, the load or no load of the third cup, the visibility or occlusion of the face, and the use of clear or opaque cups were factors in the main analyses. The analyses to determine the evolution of responses and gaze positions throughout the experiment used only the trial number as predictor. The statistical models determined main effects and first order interactions, when applicable. Only significant effects are reported in the text. Pairwise comparisons across different routines were tested with the Newman–Keuls post hoc test.

Results

Perceptual reports

Subjects reported the placing and removal of balls: they pressed “1” whenever a ball was removed and “2” whenever a new ball was placed on the table or under a cup (Fig. 2; see Methods for details). We analyzed subject performance using a logistic regression model (Hosmer–Lemeshow statistic χ2 = 2.02, area under ROC curve AUC = 0.77). Subjects’ performance in reporting the loading of the third cup was at chance level in the conditions with opaque cups (p > 0.05), and significantly improved in the transparent cups trials, when taking all the experimental trials into account (p < 10−7) (Fig. 3A). Performance was also better for simulated rather than real loads in the opaque cups (p < 10−6), due to skipped loading reports in the opaque cups condition, which impaired perceptual performance for the real loads, but not for the simulated loads (p < 0.001) (Fig. 3C). From the various sleight-of-hand maneuvers tested, the last-ball loading reports were significantly worse for the “Table” p < 0.05) and the “Stuck” (p < 0.05) conditions than for the “Standard” condition (Fig. 3A). Subjects’ performance was equivalent when the magician’s face was visible and when it was blocked (Fig. 3B).

Figure 3 Summary of subjects’ performance across the different conditions.

(a) Subjects’ performance in reporting the load of the third cup across the different routines, for the conditions with clear and opaque cups. Performance was uniform across the different routines for the opaque cups, and worse than in the conditions with clear cups (logistic regression, p < 10−7). When the cups were clear, and the load or no load of the cup was therefore visible, performance was worse for the “Table” and “Stuck” routines (logistic regression, p < 0.05). (b) Performance was similar regardless of the face being visible or not. (c) Performance was better for the “No load” condition with opaque cups (logistic regression, p < 0.001). Dashed lines show the expected chance performance level. Error bars indicate the standard error from the mean across subjects.

Subjects’ reaction times were comparable for all three cups, across the six different sleight-of-hand manipulations (for each individual condition and for the six conditions together as a whole), and for visible vs. blocked faces.

Gaze dynamics

We studied the subjects’ gaze dynamics during the viewing of each video clip (Fig. 4; see Methods for details) using a linear regression model (R2 = 0.19). Gaze distance to the third cup was highest for the “Lift” condition (p < 0.0001) and lowest for the “No ball” condition (p < 0.0001), suggesting that the “Lift” manipulation caused the largest gaze displacement (i.e. overt misdirection (Macknik et al., 2008)), whereas the “No ball” manipulation produced the smallest gaze displacement/misdirection (possibly because in the absence of a ball, subjects may allocate stronger attention to the cup) (Fig. 4A).

Figure 4 Gaze displacement from the bottom of the cup at the time of the load.

(a) Gaze distance across the different routines. The “Lift” routine caused the biggest displacement from the bottom of the cup (linear regression, p < 0.0001), while the “No ball” routine produced the smallest one (linear regression, p < 0.0001). (b) Gaze displacement was similar for the “Clear cups” and “Opaque cups” conditions. (c) Gaze displacement was similar for the “Face” and “No face” conditions. Distance is reported in degrees of visual angle, and error bars indicate the standard error from the mean across subjects.

We used a different linear regression model (R2 = 0.18) to correlate gaze distance and reaction times, and found that increased gaze distance resulted in higher reaction times (p < 0.001), with a significant effect of sleight-of-hand manipulation after controlling for the effect of gaze distance (p < 0.05).

To study the potential effect of saccadic suppression on the perceptual differences we found across conditions, we estimated the saccade production rate in the same movie frames used to measure the gaze distance to the bottom of the cup (Supplemental Figure S1). Saccade production was equivalent across the tested conditions.

Learning effects

Subjects’ performance improved over the course of the experiment (Fig. 5). In the opaque cups conditions, the number of “late findings” (i.e. ball placing reports after the magician showed the contents of the cups) decreased with trial number (logistic regression, p < 0.01 Hosmer–Lemeshow statistic χ2 = 17.23, area under ROC curve AUC = 0.92). In the transparent cups conditions, there were few “late findings”, even in the initial trials (Fig. 5A). In the clear cups conditions, correct loading reports for the third cup increased as the experiment progressed (logistic regression, p < 0.001, Hosmer–Lemeshow statistic χ2 = 15.02, area under ROC curve AUC = 0.73) (Fig. 5B). In the opaque cup conditions, subjects did not have any information about the load of the last cup, and performed at chance, therefore we found no apparent learning effect, as expected. Reaction times decreased (linear regression, p < 0.05, R2 = 0.34) (Fig. 5C) with trial number in the clear cups conditions, but remained constant in the trials with opaque cups, indicating that subjects were guessing during this condition. Gaze distance to the bottom of the third cup decreased with trial number for transparent and opaque cups (linear regression, p < 0.01, R2 = 0.31).

Figure 5 Effects of learning in perceptual reports and gaze distance.

(a) The number of late findings (placings reported after the magician shows what is under the cups) goes down with the trial number in the conditions with the opaque cups, while is very low during the experiment for the conditions with clear cups. The correlation between the trial number and the number of late findings is statistically significant in the conditions with opaque cups (logistic regression, p < 0.01). (b) Probability of subjects reporting correctly the loading of the third cup in the conditions with clear cups as a function of the trial number. The relationship is statistically significant (logistic regression, p < 0.001). (c) The reaction times of the subjects reporting the loading of the third cup (in the conditions with clear cups) decreased with the number of trials (linear regression, p < 0.05). (d) Similarly, the gaze distance (in degrees of visual angle) to the bottom of the cup decreased with the trial number (linear regression, p < 0.01). Error bars indicate standard error from the mean across subjects.

To ensure that this learning effect did not affect our other conclusions about the experimental conditions, we conducted an additional analysis of subject performance as a function of the first viewing of each condition (Supplemental Figure S2). The results are comparable to those in Fig. 3, indicating that the learning effect did not affect subject performance as a function of condition. Further, because the sequence of conditions was random and different for each subject, a systematic learning effect could not have biased our other results.

Discussion

We investigated the potential contribution of several perceptual elements in Penn & Teller’s version of the classic “Cups and balls” magic trick. We measured the perceptual performance and gaze behavior of naive observers as Teller surreptitiously introduced balls inside opaque and transparent upside down plastic cups. Contrary to the magician’s intuition, a gravity-driven drop of a ball into his hand (or to the floor) caused less misdirection, both in terms of gaze displacement and impaired perception, than alternative manipulations such as lifting the ball, or attempting to drop a ball that is stuck to the cup. Thus, perception of (the effects of gravity on) falling objects does not enhance magic misdirection, at least in the performance of this particular sleight-of-hand trick.

The contradiction between our results and the magician’s original perception may have been caused by one or more of several possible sources. One possibility is that performing the trick in a new way may have drawn his attention towards the new element (the ball dropping), and away from the common element (the loading of the cup). Successive, non-controlled repetitions of the procedure could have given the impression of a worse detection of the loading because of confirmation bias. Our results confirm that controlled experiments give valuable insight to reject (Cui et al., 2011) or accept (Otero-Millan et al., 2011) intuitive judgments about attention and misdirection formulated by magicians. Further, the three consecutive sleight-of-hand manipulations (actual or simulated loads) were presented in isolation, rather than as part of a complete “Cups and balls” magic routine (an arrangement of tricks organized in logical fashion as part of a magic performance). Finally, because an actual magician (i.e. rather than a cartoon or computer simulation) performed all maneuvers, motion features such as timing, duration, etc. could not be exactly equated across all experimental conditions. Future research using computer simulations of the magician’s sleight-of-hand movements should be conducted with the goal of replicating and generalizing the current findings to other sleights-of-hand and magic tricks.

Blocking or unblocking the magician’s face did not affect the observers’ perception or oculomotor behavior, suggesting that the “Cups and balls” magic trick does not rely on social misdirection (for instance, due to the magician’s head or eye position/movements). These results are surprising – the belief among magician’s that social misdirection, generated by the face, is one of their most powerful tools, is pervasive – though they agree with those reported by Cui et al. (2011) with a different magic trick. Together they suggest that social misdirection may differentially enhance, lessen, or fail to affect various specific magic illusions.

Also in agreement with Cui et al. (2011), we found significant effects of learning on the perception and gaze behavior of initially naive observers – the more times spectators see a trick the less effective the misdirection. Our combined results have implications for how to optimize the performance of the “Cups and balls” magic trick, and for the types of hand and object motion that maximize magic misdirection.

Supplemental Information

Supplemental Figure S1 Saccade rates during the time the cup is loaded

a) Saccade rates across the different routines. There is no appreciable difference between routines. b) Saccade rate was similar for the “Clear Cups” and “Opaque Cups” conditions. c) Saccade rate was similar for the “Face” and “No face” conditions. Error bars indicate standard error from the mean across subjects.

Click here for additional data file.

Supplemental Figure S2 Summary of subjects’ performance for the first trial of each condition

a) Subjects accuracy in reporting the load of the third cup across the different routines, for the conditions with clear and opaque cups, for the first trial of each of the 48 conditions tested. Performance is uniform across the different routines for the opaque cups, and worse than in the conditions with clear cups (logistic regression, p < 10−7). When the cups are clear, and the loading or no loading of the cup is therefore visible, performance is worse for the “Table” and “Stuck” routines (logistic regression, p < 0.05). b) Performance was similar regardless of the face being visible or not. c) Performance was better for the “No load” condition with opaque cups (logistic regression, p < 0.001). Dashed lines show the expected chance performance level. Error bars indicate the standard error from the mean across subjects.

Click here for additional data file.

Supplemental Movie S1 Video clip of the magician performing the “Standard” routine

Click here for additional data file.

Supplemental Movie S2 Video clip of the magician performing the “No ball” routine

Click here for additional data file.

Supplemental Movie S3 Video clip of the magician performing the “Lift” routine

Click here for additional data file.

Supplemental Movie S4 Video clip of the magician performing the “Table” routine

Click here for additional data file.

Supplemental Movie S5 Video clip of the magician performing the “Drop” routine

Click here for additional data file.

Supplemental Movie S6 Video clip of the magician performing the “Stuck” routine

Click here for additional data file.

We thank the NOVA scienceNOW production crew for providing the filming of the videos in the experiment, Penn & Teller for providing us with their theater, and Teller for performing the magic tricks we studied.

Additional Information and Declarations

Competing Interests

Author Contributions

Human Ethics

Susana Martinez-Conde and Stephen L. Macknik are academic editors for PeerJ.

Hector Rieiro performed the experiments, analyzed the data, contributed reagents/materials/analysis tools, wrote the paper.

Susana Martinez-Conde and Stephen L. Macknik conceived and designed the experiments, wrote the paper.

The following information was supplied relating to ethical approvals (i.e. approving body and any reference numbers):

Barrow Neurological Institute’s Internal Review Board. Protocol 04BN039.

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
