# Peer review of "Perceptual elements in Penn & Teller’s “Cups and Balls” magic trick"

_PeerJ, doi:10.7717/peerj.19_

## Round 0.1 · original submission · Major Revisions

Both reviewers requested substantive changes to the figures and methods description, and Reviewer 1 identifies several potential problem areas in the statistical analysis. In your reply letter, please explicitly identify the changes in the manuscript which address these concerns.
Reviewer 2 points out that the magician's own expectation of what elements were important for misdirection of the viewer was inconsistent with the data, which is briefly addressed in the Discussion. In my opinion, it would be worthwhile to expand this point in the discussion beyond the single paragraph in the original submitted manuscript, and clarify which perceptual or cognitive elements contribute to the expectations of the viewer, especially the naive viewer. Please note that I leave this suggestion to the discretion of the authors, and this specific point will not affect my final decision if the other issues called out by the reviewers are adequately addressed.
Please ensure that the revised manuscript conforms to the PeerJ policy regarding data and materials sharing:
Data and Materials Sharing
1 PeerJ is committed to improving scholarly communications and as part of this commitment, all authors are responsible for making materials, data and associated protocols available to readers without delay. The preferred way to meet this requirement is to publicly deposit as noted below. Cases of non-compliance will be investigated by PeerJ, which reserves the right to act on the results of the investigation.
...
- Where suitable domain-specific repositories do not exist, authors may deposit in either Dryad or an institutional repository and provide the access information with the manuscript. Alternately, authors may choose to deposit non-standard data (including figures, posters, rich media) on Figshare or PeerJ PrePrints, for example. In all cases, the DOI reference (where applicable) should be provided in the article.
- Any supporting data sets for which there are no suitable repositories must be made available as publishable Supplemental Information files by PeerJ.
...

The full policy is viewable in Policy and Procedures page at: https://peerj.com/about/policies-and-procedures/#data-materials-sharing.

Reviewer 1 ·

Basic reporting

No comments.

Experimental design

A lot of the points I make with regard to "Experimental Design" have flow on effects to the "Validity of the Findings". I keep them together in one section for clarity.

Methods & Results:
• Please provide more detail as to the experimental equipment setup. What was the resolution of the stimulus (in pixels and in degrees of visual angle), What was the luminance, and contrast of the stimulus? Was the experiment conducted in a controlled environment (dark, quiet?).
• Eye movements: You note that blink periods were removed. Were the subjects told to fixate on a particular part of the stimulus? If so then, how can you generalise your findings to a natural viewing of the magic trick. If not, then given that saccadic suppression is a very well known and studied phenomenon did you attempt to assess whether the key frames of the video that contained the manipulation occurred during a saccadic eye movement?
• Given that one of your measures of interest was reaction time, were removals (button 1) and placings (button 2) equally easy to press?
• How was the order of stimuli determined? If it was random was the same order used for each subject? This has profound implications for the validity of your learning findings.
• Data analysis. Please provide more details about the results of your data analysis. For regression analyses please provide measures of goodness of fit.
• Given that you are using the Newman-Kuels method to do pair-wise comparisons, please explain how you are controlling for the increasing experiment-wise alpha level given the number of independent variables you are testing.
• You mention that your “statistical models determined main effects and first order interactions” but you don’t report these anywhere.
• In several places throughout the paper you mention performance without qualifying which of the several measures of performance you tested you are referring to.
• Given that you suggest that there was a learning effect how did you deal with this in the main analysis. Did you average all trials? Naïve trials (such as would be most relevant to a person seeing a trick for the first time)?, learned trials?
Figures
• Please be consistent with your error-bars, you switch from 95% confidence intervals (figure 2) to SEM (Figures 3-4) and this can lead to a cursory reader assuming that later figures have more significance than they actually do.
• Figure 2ABC) Please indicate where chance performance would be on these figures. Are the subjects worse than chance when tested with opaque cups?
• Figure 3. Please indicate in the figure caption that dva refers to degrees of visual angle? or change the axis label to reflect this.
• Figure4a. Please indicate what number the # of late findings is in relation to.
• Please explain why you don't present the graph for opaque cups in Figure 4B&C.
• I am concerned that given the findings concerning your other variables, comparisons of trial number to your dependent variables are potentially confounded by order effects. Without knowing how the conditions were ordered it is not possible to evaluate the data being presented in Figure 4.

Validity of the findings

See my comments in the "Experimental Design" section.

·

Basic reporting

The scientific part of the paper is well-written, but I found the non-scientific part, namely the description of the the performance of the magic, difficult to comprehend. Maybe I am simply not familiar with stage magic but it took me a long time to get a sense of what the magician actually did. The inclusion of a video clip will greatly improve the readability of the paper. If that is not possible due to copyright or technical issues, I think the introduction needs to be revised to give more detailed and precise description of the performance. Since I was not familiar with Cups and Balls, I turned to youtube, hoping to see how Penn and Teller perform the Cups and Balls magic. This probably was not such a good idea because Cups and Balls, as performed in Penn and Teller’s various stage shows and television programs, appeared to be not exactly the same Cups and Balls magic studied in the paper (in the youtube clips, the magician does not tilt the transparent cup to make the ball fall into his hand). This is a source of confusion that can be avoided if the introduction provides more background information.

The second paragraph of Introduction describes how the magic was invented. According to the description, the magician was surprised by how effective the illusion was. The falling of the ball distracted the attention of the magician such that he himself did not see the loading of the cup. However, Figure 2 clearly shows that the subjects could reliably detect the loading of the cup, thus showing that the magic was not magical, after all! How can a successful magician be so wrong about the effectiveness of his performance? I’m puzzled. The description of the magician’s invention of the magic, rather than helping to motivate the experiment, only confused this reviewer.

Experimental design

The design of the experiment is appropriate for the research question. I have a few suggestions on the presentation:

1. The description of the experiment lacks some information: a) the running length of the video; b) randomization of the conditions, if any; and c) The exact instruction that was given to the subject.

2. In “Data analysis” section, “placed or removed” was written as “placed OUR removed”.

3. The Results section begins with a summary figure (Figure 2). This is inadequate. I’d like to see how the subject actually performed in the detection task. A few examples of “raw” data (time points of button press in relationship to the time points where the magician placed or removed the balls) should be provided.

4. The subjects were asked to detect two events (removal and placing of the ball) but Figure 2A collapsed them into one single measure (probability of correct report). The analysis should separate the two, because the two events have different significance. The stated goal of the experiment was to determine “whether the falling ball in Penn & Teller's Cups and Balls generated stronger misdirection, as hypothesized by Teller, than alternative manipulations.” In this context, the “loading” of the cup (ie. placement of the ball) more directly reflects the effectiveness of the illusion than the removal of the ball. Recall that in the introduction, the magician was said to be surprised that he did not see the load, thus making the load the more important. Either that, or I have misunderstood the research question.

If the separating Figure 2A into two turns out to be less than insightful, the authors should at least briefly report the results or make a comment on it.

Validity of the findings

The research question is: how effective was the illusion? Figure 2 answers the question: the falling of the ball was not as effective as the magician suspected. However, the authors found a learning effect. The more the subjects viewed the magic, the less effective the illusion was. Given that, the interpretation of Figure 2A becomes complicated. It is still possible that the magician's manipulation was effective, but only for the first time. This question is not addressed in the manuscript. The manuscript therefore fails to give a compelling answer to the research question expressed in Introduction.

This unsatisfactory situation is difficult to avoid given the subject matter. I don't think it makes the findings invalid but the authors should either try to see if it can be addressed in some way from the experimental data, express the research question differently, or acknowledge the problem in discussion.

A second issue is with Figure 2C. The effect is acknowledged in the text but its significance is never commented upon. Since it is difficult to detect the load in the opaque cup condition, how can the no load condition affect the results? What does this mean?

---

## Round 0.2 · Minor Revisions

Thank you for your thorough response to the points raised in the previous reviews. I believe this manuscript will be acceptable, pending amendment/explanation of the point raised by Reviewer 1 regarding the statistical values in the Results section of the amended manuscript.

Reviewer 1 ·

Basic reporting

I am satisfied that the author's of this manuscript have addressed my earlier concerns.

Experimental design

I am satisfied that the author's of this manuscript have addressed my earlier concerns.

Validity of the findings

I am satisfied that the author's of this manuscript have addressed my earlier concerns.

I want to point out that in the Results-> Perceptual Reports section, all of the chi-square and AUC values are identical despite different significance levels. I am not sure if this is intended, if not the author's should consider correcting the values prior to publication.

---

## Round 0.3 · accepted · Accept

Congratulations, and I appreciate the quick turnaround on the most recent version. I look forward to seeing the published document.